# Social Touch Reduces Pain Perception—An fMRI Study of Cortical Mechanisms

**DOI:** 10.3390/brainsci13030393

**Published:** 2023-02-24

**Authors:** Mattias Savallampi, Anne M. S. Maallo, Sumaiya Shaikh, Francis McGlone, Frédérique J. Bariguian-Revel, Håkan Olausson, Rebecca Boehme

**Affiliations:** 1Center for Social and Affective Neuroscience, Department of Biomedical and Clinical Sciences, Linköping University, 58185 Linköping, Sweden; 2Research Centre Brain & Behavior, Liverpool John Moores University, Liverpool L3 5UZ, UK; 3Haleon, GSK Consumer Healthcare S.A, 1260 Nyon, Switzerland; 4Department of Clinical Neurophysiology, Linköping University Hospital, 58185 Linköping, Sweden; 5Center for Medical Imaging and Visualization, Linköping University, 58185 Linköping, Sweden

**Keywords:** social touch, pain, fMRI, PAG, insula

## Abstract

Unmyelinated low-threshold mechanoreceptors (C-tactile, CT) in the human skin are important for signaling information about hedonic aspects of touch. We have previously reported that CT-targeted brush stroking by means of a robot reduces experimental mechanical pain. To improve the ecological validity of the stimulation, we developed standardized human–human touch gestures for signaling attention and calming. The attention gesture is characterized by tapping of the skin and is perceived as neither pleasant nor unpleasant, i.e., neutral. The calming gesture is characterized by slow stroking of the skin and is perceived as moderately to very pleasant. Furthermore, the attention (tapping) gesture is ineffective, whereas the calming (stroking) gesture is effective in activating CT-afferents. We conducted an fMRI study (*n* = 32) and capitalized on the previous development of touch gestures. We also developed an MR compatible stimulator for high-precision mechanical pain stimulation of the thenar region of the hand. Skin-to-skin touching (stroking or tapping) was applied and was followed by low and high pain. When the stroking gesture preceded pain, the pain was rated as less intense. When the tapping gesture preceded the pain, the pain was rated as more intense. Individual pain perception related to insula activation, but the activation was not higher for stroking than for tapping in any brain area during the stimulation period. However, during the evaluation period, stronger activation in the periaqueductal gray matter was observed after calming touch compared to after tapping touch. This finding invites speculation that human–human gentle skin stroking, effective in activating CT-afferents, reduced pain through neural processes involving CT-afferents and the descending pain pathway.

## 1. Introduction

The perception of pain depends on a myriad of factors, broadly summarized as top-down and bottom-up. Top-down factors include states such as mood, expectation, context, resilience, catastrophizing, prior experience etc., and bottom-up factors include signals from the area of the body where the painful stimulus occurred, prior history of injury, and, importantly, the level of activity in nociceptors as well as in non-nociceptive sensory nerves that code for touch, temperature, and itch. This interdependent complexity of pain was first recognized in 1965 [1]. If not for this gate control theory, pain perception might still be associated with the intensity of the pain stimulus and the degree of damage caused to the affected tissue and not with the modulating properties of the central nervous system.

The gate control theory provides an explanation of why pain relief can be obtained by rubbing or massaging an injured or a painful area. The gate control theory proposes that activation of myelinated mechanoreceptive (Aβ) afferents at the site of an injury activates inhibitory interneurons in the spinal cord, which in turn decrease the amount of nociceptive information signaled to the brain. At the time when this theory was presented, human touch was thought to be signaled exclusively by Aβ afferents. However, since for about 20 years, it has been known that the human skin is also innervated by unmyelinated low-threshold fibers (C-tactile, CT) that respond to gentle touch [2]. Whereas Aβ afferents respond preferentially to rapid skin deformations, CT afferents respond optimally to slow stroking touch with a preferred velocity of 1–10 cm/s [3].

Indeed, CT-targeted slowly stroking touch in healthy subjects causes a robust short-lasting relief of experimental pain [4,5,6,7], evident as a 10% reduction on a visual analogue scale. The analgesic effect is present already in newborns [8] and seems to depend on intact CT afferents [9]. However, the social aspect of the touch interaction and the emotional relationship also play a role, for example, handholding, which is less effective in activating CT-afferents, has been shown to reduce experimental pain as well [10,11].

Stroking the skin of a person’s forearm with a soft brush is not a naturalistic or ecologically valid way of reducing pain. Therefore, in this study, we investigated the effects of human-to-human touch on experimental pain. The experimenter used their hand to produce two types of gestures that consisted of tapping or slowly stroking of the subject’s forearm. Using intraneural recordings with the microneurography technique, we have previously shown that the tapping gesture is an effective stimulus for activating Aβ afferents, whereas the slowly stroking gestures is effective in activating CT afferents [12]. Furthermore, the tapping gesture is intuitively understood by the receiver as a signal for attention, whereas the slowly stroking gesture is intuitively understood as a signal for calming or love [13,14].

Social touch is known to play a crucial role in development, bonding, and social interactions [15]. Slow, affective tactile stimulation using soft brushes reliably activates the insula [16], an area otherwise involved in perceiving signals from within the own body (interoception) [17,18] including pain [19]. Additionally, more ecological types of interpersonal touch involving skin-to-skin stroking evokes insula activity [20]. The important role of social touch during development and bonding is mediated in part via the oxytocin system, which is reliably activated in positive, affective social touch interactions [21,22,23,24]. This relationship seems to be mediated by connections of the periaqueductal gray (PAG) matter to oxytocin neurons [25]. Interestingly, the PAG is also well known to play a crucial role in the descending pain system and has been implicated in pain analgesia [10,26]. PAG receives inputs from the anterior cingulate cortex (ACC), an area known to play a role fewn higher level functions pertaining to pain processing [27], e.g., threat prediction, but also involved in encoding self-referential aspects of cognition [28,29], even during social touch [20].

Here, we studied touch analgesia using functional brain imaging. We hypothesized that the slowly stroking gesture, effective in activating CT afferents, would reduce the intensity of the experienced pain. The attention gesture, ineffective in activating CT afferents, would be less effective. We also hypothesized that the slowly stroking gesture will reduce pain through a mechanism involving insular cortex [30] and periaqueductal gray matter [25].

## 2. Materials and Methods

### 2.1. Participants

Thirty-two participants were included. All participants were interviewed and reported to be healthy. Exclusion criteria were any MR contraindications, any general health concern, and a history of psychiatric illness or of chronic pain. For screening purposes, the participants filled out the following questionnaires: McGill Pain Questionnaire, Beck’s Depression Inventory, and State-Trait Anxiety Inventory. The National Ethics Authority approved the study (dnr 2017/598-31) and written informed consent was obtained after study description.

### 2.2. Procedures

During functional magnetic resonance imaging (fMRI), participants performed a touch–pain interaction task. In this task, participants received high- and low-level pressure stimuli after two types of touch (see details below). They were introduced to the task before entering the scanner room and their individual pain threshold was determined. Pressure was delivered using a MR-safe pressure device built by the lab. It consists of a pneumatic pump, which uses air pressure to push a cylindrical 1 cm^2^ plastic peg into the thenar muscle at the base of the thumb (Appendix A).

For threshold determination, the participant sat in a chair and placed their hand in the pressure device positioned on a table. They were instructed to press a button on a laptop in front of them when the sensation became clearly painful. The pump slowly increased the pressure until the participant pressed the button, upon which air pressure was released immediately. This procedure was repeated three times with three seconds rest in between. In total, 10% was added and 50% subtracted from the average value of these three measurements. These values were then used for “high pressure” and “low pressure” stimuli during imaging. To ensure participants’ safety and good data quality (as pain might cause movement during fMRI), the pressure stimulus was tested once more before the experiment started. Participants rated the perceived pain on a scale ranging from 0 to 10. In case they rated higher than 3, the threshold calibration was repeated. 

After threshold calibration, participants were placed in the fMRI scanner. The device was fixed to their left hand, which they rested on their upper body. In their right hand, participants held a button box and were informed to use their index and middle fingers to report pain ratings when prompted during the task. They were presented with a computer screen through googles, which showed a white fixation cross on black background during the task and a question (“How did this feel?”) and the scale (0–100) during the rating period. The participants did not see numerical values but just the endpoints “no pain” and “high pain” on a scale with a sliding cursor, which could be moved using the buttons.

### 2.3. Touch–Pain Interaction Task

The touch–pain interaction task (Figure 1) consisted of four blocks of approximately 7 min each, resulting in a total length of 28 min. During each block, the participants received a total of 6 stimuli (3× low pressure and 3× high pressure in a randomized order). These stimuli were preceded by 30 s of two different types of touching on the left forearm: a tapping, which gives as weak activation of CT afferents, and a slow stroking, which gives a strong activation of CT afferents. The tapping and stroking stimuli were delivered by an experimenter who had trained extensively to perform the gestures in a consistent manner [13]. Per block, only one type of touch was administered to reduce spill-over effects of the different touch types into the next trial, resulting in two tapping-blocks and two stroking-blocks. The order of blocks was randomized across participants. In between blocks, there was a short break, where the scanner conductor checked in with the participant to ensure compliance and safety and the participants had a short resting period (approximately 60 s). Duration, timing, and location of the touch was based on pilot testing and the touch and pain task commonly used in our lab [13,20,31,32]; we did, however, not systematically identify the most effective timing and distance between touch and pain stimuli. The touch was administered by a female nurse who greeted the participants shortly before the experiment started. During the task, she stood next to the scanner bore and received auditory cues on the timing and type of touch via headphones.

High- and low-level pressures were delivered at the beforehand determined individual values. The pump started pumping up to this value after the touch finished and held at the target value for 5 s before it released the air pressure. After each pain stimulus, a rating scale appeared on the screen and participants were asked to rate the perceived pain using the buttons on the box in their right hand. Participants had 14 s to answer. 

### 2.4. Brain Imaging

The MRI session consisted of an anatomical T1 scan and four functional runs during the touch–pain interaction task. The images were acquired using a 3 T Siemens Prisma scanner with a 64-channel head coil. T1 images were acquired with the following settings: repetition time = 2300 ms; echo time = 2.36 ms; flip angle = 8°; field of view = 288 × 288 mm^2^; voxel resolution = 0.87 × 0.87 × 0.90 mm^3^. Functional T2-weighted echo-planar images (EPI) were collected using the following settings: 48 multiband slices (3 slices measured simultaneously), TR = 1030 ms, TE = 30 ms, slice thickness = 3 mm, matrix size = 64 × 64, field of view = 192 × 192 mm^2^, in-plane voxel resolution = 3 mm^2^, flip angle = 63°.

### 2.5. Statistical Analysis

Data cannot be made publicly available due to missing participant consent. Questionnaire data (except state anxiety) and rating data were not normally distributed (Shapiro–Wilk test for low pressure ratings: 0.92, *p* = 0.028, for high pressure ratings: 0.91, *p* = 0.016) and analyzed in SPSS (IBM) using non-parametric tests (Friedman test, Wilcoxon signed rank test). Functional MRI data were analyzed using statistical parametric mapping (SPM12, Wellcome Department of Imaging Neuroscience, London, UK; http://www.fil.ion.ucl.ac.uk/spm, accessed on 21 February 2023) in Matlab R2018b (The MathWorks, Natick, MA, USA). The following steps were performed for preprocessing: motion correction, co-registration of the mean EPI to the T1, spatial normalization to the MNI template, and segmentation of the T1 image using the unified segmentation approach [33]. Normalization parameters were applied to all EPIs. EPIs were spatially smoothed with an isotropic Gaussian kernel (6 mm full width at half maximum). For statistical analysis of the blood-oxygen-level-dependent (BOLD) response, the general linear model approach was used as implemented in SPM12.

Three first level models were implemented (Figure 1). (1) To test for general effects of the two different touch types and the two different pain types, regressors of interests were the pain types (high and low pain), the touch type (tapping and stroking), and the rating phase. (2) To test for the individual modulation of the signal based on pain perception, ratings given after each trial were included as a parametric modulator on the stimulation period (from the start of the touch to the end of the pain stimulation). (3) Based on our previous finding in fibromyalgia, where the pain rating phase showed the main effect [32], we were additionally interested in the individual modulation of the signal during the rating phase. Therefore, in the third model, ratings were included as parametric modulator on the rating phase.

In all models, realignment parameters were included as regressors-of-no-interest to account for movement associated variance. All regressors were convolved with the hemodynamic response function. Due to the short TR, the FAST option [34] was used to improve autocorrelation modeling performance [35]. The resulting beta parameter estimate maps were then analyzed at the group level. Repeated measures ANOVA (first level model 1) and paired t-tests (first level models 2 and 3) were used to compare conditions. Family-wise error (FWE) correction at the voxel level was used to correct for multiple comparisons at the whole-brain level and for small volume correction (SVC) based on our a priori regions of interest (ROI): insula, as the insula is implicated in the processing of affective touch [16], pain [19], and the awareness of feelings from the body [17,18], periaqueductal gray (PAG) matter, which plays an important role in the descending pain pathway [26], and the anterior cingulate cortex (ACC), known to contribute to PAG regulation [27]. Insula and ACC ROIs were anatomically defined [36] and the PAG ROI was defined as a 4 mm sphere around the coordinates [−4 −29 −12] based on a recent meta-analysis [26]. Graphics were made in JASP 0.16 and MRIcron 1.4.

## 3. Results

### 3.1. Demographics

Both state and trait anxiety were in the low range (state anxiety median = 29 (IQR 8); trait anxiety mean = 35.2 ± 8.5, Table 1). Depression scores were mostly in the normal range (median 3 (IQR 5), range 0–17). Two people scored in the range of mild depressive symptoms. Current pain was rated on the McGill scale with a median of 0 (IQR 8), range 0–29, which is at the low end of the scale’s range (0–78).

### 3.2. Behavior

During the touch–pain interaction task, there was an overall statistically significant difference in perceived pain depending on type of pressure and type of touch (χ^2^(3) = 50.6, *p* < 0.001). Post-hoc comparisons revealed that the pain ratings during high pressure were significantly higher than during low pressure (Figure 2, median high: 27.92 (IQR 31.5), range 3.6–70.8; median low: 8.25 (IQR 18.49), range 0–41.5; Z = −4.36, *p* < 0.001, effect size: 0.9). Overall, stroking preceding the high/low pressure-stimulus led to lower pain ratings than tapping (median stroking: 16.5 (IQR 24.8), median tapping: 22.75 (IQR 23.42), Z = −2.35, *p* = 0.019, effect size: 0.2). This effect was mostly driven by the individual differences in pain ratings when the low-pressure stimuli were preceded by stroking or tapping touch (median stroking: 8.5 (IQR 16.8), mean stroking: 10.6 ± 10.2; median tapping: 8.2 IQR 20.2, mean tapping: 13.6 ± 13.2; Z = −2.76, *p* = 0.006, effect size: 0.2). While not significant, pain ratings were also lower when the high-pressure stimuli were preceded with stroking compared to tapping (median stroking: 25.5 (IQR 28.1), mean stroking 29.76 ± 21.4, median tapping: 27.83 (IQR31.9), mean tapping: 31.9 ± 21.8, Z = −1.58, *p* = 0.113).

### 3.3. Brain Imaging

The main effect of touch (both types) to the left forearm was located bilaterally in the posterior insula (right: [48 −31 23], t = 11.09, *p* < 0.001; left: [−51 −28 20], t = 6.7, *p* < 0.001; both FWE-corrected for the whole brain, Figure 3) and in the right somatosensory cortex ([21 −20 65], t = 5.9, *p* < 0.001, FWE-corrected for the whole brain). Additional peaks (FWE-corrected for the whole brain) were found in right superior temporal gyrus ([69 −40 17], t = 7.72, *p* < 0.001, [63 −55 20], t = 6.5, *p* < 0.001) and left middle temporal gyrus ([−54 −64 17], t = 6.35, *p* = 0.001, [−48 −58 8], t = 4.9, *p* = 0.021). Tapping touch elicited significantly higher activation than stroking touch in the cerebellum (Table 2), which potentially reflects movement preparation, as the tapping touch can be understood as an attention seeking gesture [13]. There was no higher activation for stroking than for tapping in any area.

The main effect of pressure (both high and low intensity) included many regions in the frontal and temporal cortices, as well as somatosensory cortices, cingulate, and insula (Table 3, Figure 4).

There was no significant difference between low- and high-pressure stimuli when correcting for multiple comparisons at the whole-brain level. Furthermore, there was no significant interactions between the two pressure stimuli with the two types of preceding touch. However, this type of analysis did not take individual pain perception into account.

In the second analysis, we included individual pain ratings as parametric modulator in the analysis. This was done to account for the subjective quality of pain perception. The parametric modulator explains variance on top of the variance modeled by the regressors. Since we did not find a difference in BOLD signal between high- and low-pressure stimuli (see above), we pooled high- and low-pressure trials in this model. This approach revealed that brain activity in the bilateral insula (ROI analysis) was related to the subjective pain level ratings (right: [39 −1 11] t = 3.9, *p*(SVC) = 0.028; left: [−39 −1 −10] t = 3.73, *p*(SVC) = 0.046, [−42 −4 8] t = 3.52, *p*(SVC) = 0.047, [−45 −7 −7] t = 3.71, *p*(SVC) = 0.049, Figure 5). This was independent of the type of preceding touch. There was no significant co-variation of signal strength with individual ratings in the two other ROIs, ACC and PAG. 

In a third exploratory analysis, we added the individual ratings as a parametric modulator to the rating period [32], i.e., during the period when participants were evaluating their percept. Interestingly, here, we found no significant covariation of BOLD signal with the ratings. We then compared the rating periods after calming touch with the rating periods after tapping touch (i.e., after removing variance related to individual ratings) and found more activation after calming in the PAG (ROI analysis) ([−3 −28 −16], t = 4.1, *p*(SVC) = 0.001, Figure 6), in the right anterior insula (ROI analysis) ([42 11 −4], t = 4.2, *p*(SVC) = 0.032), and in the ACC (ROI analysis) (left: [0 17 29], t = 5.32, *p* = 0.002, [−6 20 29], t = 5.31, *p* = 0.002; right: [3 20 26], t = 5.35, *p* = 0.001).

## 4. Discussion

Using naturalistic human-to-human (skin-to-skin) touch in combination with experimental cutaneous and muscle pain, we demonstrated that a slowly stroking gesture reduced experimental pain in a low pain condition. In contrast, a tapping gesture did not reduce experimental pain to the same extent. In our experimental design, the touch gestures preceded pain. Individual pain perception was related to insula activity during the stimulation period. However, the activation was not higher for stroking than for tapping in any brain area during the stimulation period.

The slow stroking is expected to activate CT afferents, whereas the tapping gesture is not [37]. However, both touch types activated the insula cortex bilaterally—there was no significant difference in the BOLD signal in the insula, as has been reported previously when applying slow and fast brushing on the arm [16]. The main difference in the stimulation here was that both touch types were skin-to-skin touch, not delivered by a tool. This might suggest that the social aspects (human touch, context, intention) play a more important role regarding insula activity than the speed of stimulation or activation of certain receptor types in the periphery. We have shown before that the slowly stroking gesture is intuitively understood as a signal for calm or love, whereas the tapping gesture is intuitively understood as a signal for attention [13]. However, here, we did not ask participants for an interpretation of the two types of touch. The activation of the cerebellum in relation to the tapping touch supports this further as it might indicate motor preparation.

We found that insula cortex activity during the stimulation phase was associated with the individually reported level of pain intensity. This is in line with previous reports implicating the insula as a key area for pain perception [19,38,39]. Notably, the BOLD signal in the insula related to the individual ratings only during the stimulation period but not during the actual rating period. The reduction in pain after stroking compared to tapping was only significant in the low-pressure condition in our set-up; however, the effect that CT-targeted touch attenuates pain perception has been described extensively before [4,5,6,7,8,9].

An exploratory analysis found that during the rating period PAG activity was increased following the slow stroking stimulation compared to the tapping touch. PAG is considered an important regulatory area of the descending pain processing pathway. It has anatomical connections to laminae V, VII, and VIII of the dorsal horn of the spinal cord, and might through this pathway downregulate pain perception [26]. It is also an important relay station of the ascending pain pathway, receiving inputs from spinal cord laminae V, VII, and X. Lesions in the PAG are associated with an increase in pain conditions [40], and PAG contains a large numbers of opioid receptors—a potential mechanism for pain analgesia [26,41]. Overall, there is good evidence, that opioidergic mechanisms through the descending pain pathway including the PAG underlies analgesic effects [42,43]. Interestingly, our other regions of interest, insula and ACC, also showed more activity during the evaluation period after slow stroking compared to tapping, which might reflect their involvement in modulation of subjective pain percepts through PAG.

Distraction from a painful stimulus reduces perceived pain intensity through a mechanism which also involves the PAG [44]. In our design, one could understand the tapping as more distracting than the slow stroking touch, and we have shown before that it is understood as an attention-seeking gesture [14]. We did, however, find more activation of PAG during pain evaluation after the slow stroking than after the tapping touch. This might be due to the PAG’s known involvement in positive affect: the PAG is activated when experiencing love [45,46] and when listening to music that evokes a feeling of shivering down the spine [47]. These findings might explain the PAG activation after the slow stroking gesture, which we have previously shown to be interpreted as a gesture of calming or love [14]. However, also aversive emotional experiences, e.g., aversive sounds activate the PAG [48]. PAG is also implicated in the processing of affective social touch: recent animal work demonstrates that the PAG plays a crucial role in mediating the long-term preference of positive tactile stimulation through activation of oxytocin neurons in paraventricular hypothalamus via a dipeptidergic excitatory circuit [25]. We did not find PAG activation in response to slow stroking after the actual stimulation, but instead during the evaluation of pain. This could be interpreted as a secondary downregulation of the pain perception after the actual pain experience, potentially related to the evaluation process. PAG has been reported to be involved in placebo analgesia as well [26]. We speculate that the slow stroking touch might evoke a placebo-like state leading to pain analgesia during the evaluation of pain.

There are several open questions that future lines of research need to investigate in more detail. First, our sample was too small to investigate gender- or age-related effects, which might be of special importance when considering potential applications that would utilize social touch for analgesia in pain conditions. The role of social touch in pain analgesia and the potential involvement of different neuroendocrine systems warrant further investigation, specifically with regard to potential applications in chronic pain conditions. There are ongoing efforts to develop CT-derived analgesic drugs based on transgenic mouse models that suggest that pain relief to CT-targeted touch is mediated through spinal cord TAFA4 release [49]. Our demonstration of a coupling between ecologically relevant touch, insular-PAG-related brain network activation, and reduction of experimental pain is consistent with a view where human pain relief is mediated through a mechanism where positive social touch induces a pain-protective emotional state. Whether there is a role for TAFA4 in touch-induced human analgesia remains to be investigated.

## 5. Conclusions

In conclusion, in this project, we have an identified non-invasive social somatosensory stimulus that reduces experimental pain. The cortical mechanisms of pain-relieving effects are not fully understood. Based on the current findings, we speculate that descending signaling induced a pain-resilient emotional state through mechanisms mediated by the insular cortex and the PAG.

## Figures and Tables

**Figure 1 brainsci-13-00393-f001:**
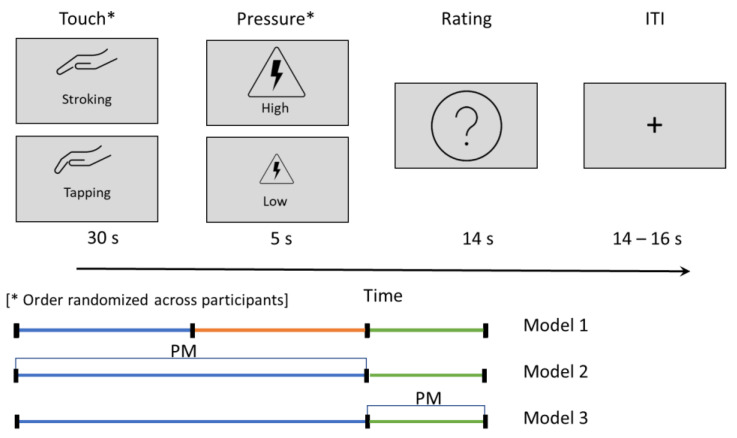
Touch–pain interaction task. Touch types were randomized across four blocks. High- and low-pressure stimuli were randomized within each block. Intertrial interval (ITI) was jittered. The trial phases modeled in the three fMRI first level models are indicated in the bottom. PM = parametric modulator.

**Figure 2 brainsci-13-00393-f002:**
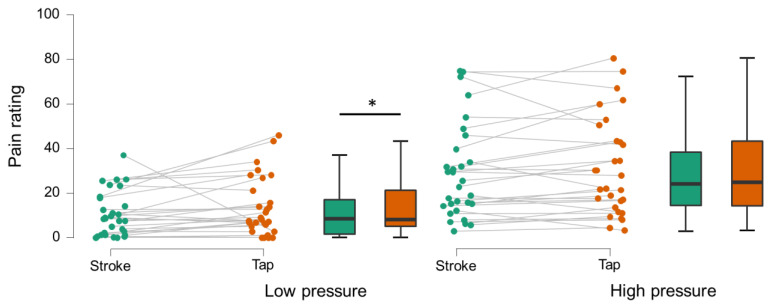
Pain ratings for low- and high-pressure trials. Dots display individual averages of ratings for low-pressure (**left**) and high-pressure (**right**) trials. Stroking (green) reduced pain ratings of low-pressure trials significantly (Z = −2.76, *p* = 0.006) compared to tapping (orange). Boxplots display median (center bar), first–third quartile (box), and 1.5 IQR (whiskers); * indicates significant difference.

**Figure 3 brainsci-13-00393-f003:**
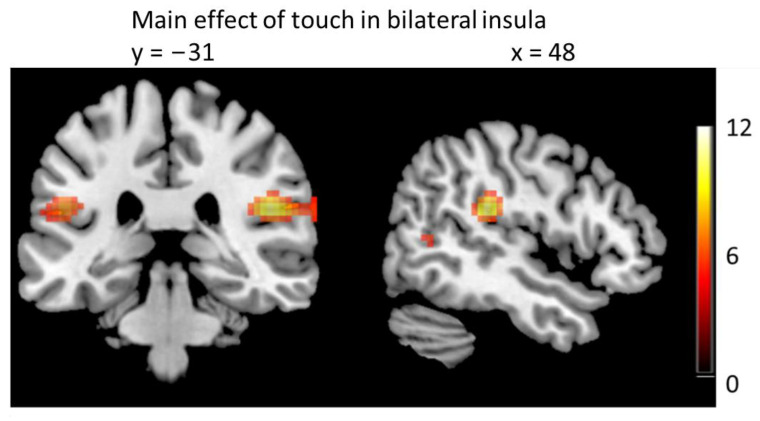
Stroking and tapping touch led to significant activation in bilateral posterior insula. Color bar indicates t-values. Thresholded at *p* < 0.05, FWE-corrected for the whole brain.

**Figure 4 brainsci-13-00393-f004:**
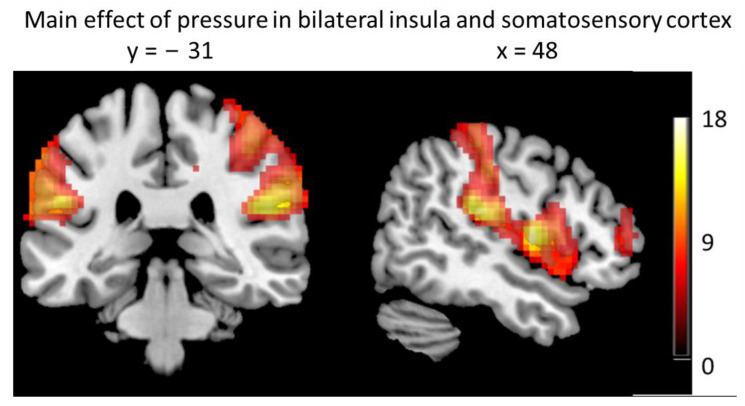
Main effect of pressure stimulus (high and low combined), thresholded at *p* = 0.05, FWE-corrected at the whole brain level, cluster-size > 50.

**Figure 5 brainsci-13-00393-f005:**
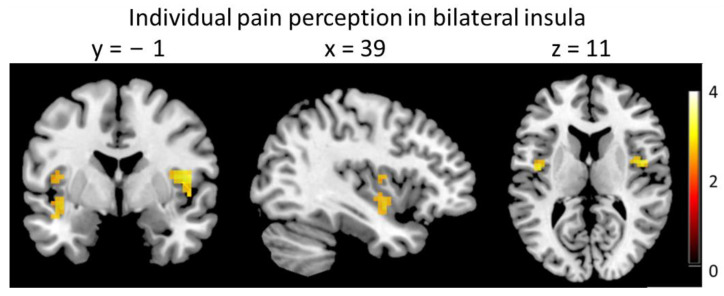
Individual pain perception relates to insula activity during the stimulation period, *p* < 0.001, clusters-size > 10 for display purpose.

**Figure 6 brainsci-13-00393-f006:**
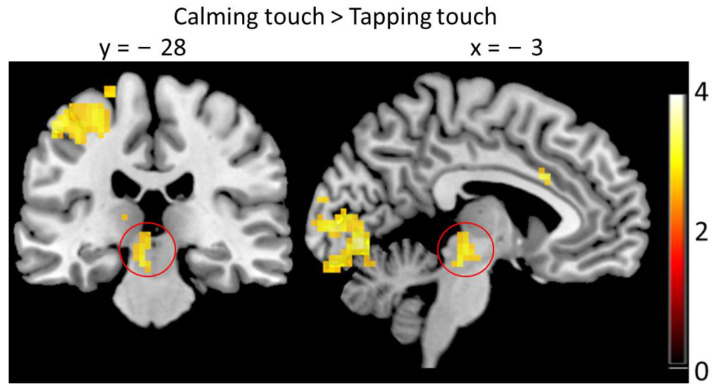
More activation in the PAG after calming touch than after tapping touch during the rating period (red circle), *p* < 0.001, clusters-size > 10 for display purpose. Additionally, visible clusters are not significant at the whole brain level and are not part of the ROI.

**Table 1 brainsci-13-00393-t001:** Participant characteristics.

Age	23.5 (IQR 9.75)
Sex	13 male, 19 female
State-Trait Anxiety Inventory	29 (IQR 8) 35.2 ± 8.5
Beck’s Depression Inventory	3 (IQR 5)
McGill Pain Questionnaire	0 (IQR 8)

**Table 2 brainsci-13-00393-t002:** Right cerebellar areas showing significantly more activation during tapping than during stroking.

Area	X	Y	Z	Clustersize	T	p (FWE-corr)
Anterior lobe	15	−40	−25	128	7.27	<0.001
Culmen	9	−49	−16		6.70	<0.001
Declive	6	−64	−19		6.68	<0.001
Posterior Lobe	15	−61	−19		5.38	0.03
Inferior Semi-Lunar Lobule	15	−73	−37	13	6.04	<0.001

**Table 3 brainsci-13-00393-t003:** Main effect of low and high pressure combined. R = right, L = left, FWE-corr = family-wise-error-corrected.

Area	Hemisphere	Cluster-size	p (FWE-corr)	T	X	Y	Z
Insula	R	2912	<0.001	19.36	42	−1	11
				17.39	51	−28	20
				12.81	36	14	5
				11.87	33	23	2
				11.84	39	−19	17
				11.60	42	11	−4
				11.51	42	5	−7
Postcentral Gyrus			<0.001	17.70	54	−25	23
				14.26	54	−16	17
				13.79	54	−19	38
				9.95	45	−28	53
				9.88	42	−31	56
Precentral Gyrus			<0.001	14.28	54	8	8
Inferior Frontal Gyrus			<0.001	13.30	39	17	−7
				13.19	57	11	23
				12.91	60	8	29
Inferior Parietal Lobule	L	1357	<0.001	16.38	−63	−25	23
				16.19	−60	−22	29
Insula			<0.001	13.98	−39	−1	−7
				13.95	−42	−4	2
				13.63	−45	−7	8
				13.36	−33	14	2
				12.89	−39	−4	14
				9.54	−30	23	8
Postcentral Gyrus	L		<0.001	13.27	−63	−22	38
Superior Temporal Gyrus			<0.001	12.29	−51	−4	5
				8.53	−54	5	−1
Precentral Gyrus			<0.001	9.69	−54	2	8
Inferior Frontal Gyrus			<0.001	9.35	−57	8	23
				9.23	−57	5	17
Supramarginal Gyrus			0.01	5.33	−66	−43	29
Lentiform Nucleus	R	110	<0.001	12.18	12	8	−4
Parahippocampal Gyrus			<0.001	7.41	18	−1	−10
Lateral Globus Pallidus	L	64	<0.001	8.61	−9	5	−4
Inferior Frontal Gyrus	R	87	<0.001	7.88	45	41	5
Middle Frontal Gyrus			<0.001	5.62	45	44	20

## Data Availability

Data cannot be shared publicly due to missing consent by the participants.

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
