# Peer review of "Social Touch Reduces Pain Perception—An fMRI Study of Cortical Mechanisms"

_brainsci, 2023, doi:10.3390/brainsci13030393_

Round 1

Reviewer 1 Report

In this study, human-human gentle skin stroking is effective in activating CT afferents and reducing pain which is most likely through neural processes of the descending pain pathway. The researchers developed standardized human-human touch gestures for signaling attention and calming. The attention gesture was by tapping and the calming gesture was by slow stroking of the skin, where the attention gesture was ineffective and calming gesture effective in activating CT-afferents. The investigators studied touch analgesia using functional brain imaging. The slow stroking gesture reduces pain through a mechanism involving the insular cortex and periaqueductal gray. This study showed that non-invasive social somatosensory stimulus could reduce experimental pain. The study group was all healthy volunteers with no pain. Based on the findings, the researchers propose that the insular cortex and the PAG are involved. However, other for other potential cortical mechanisms further studies are required.

This is a valuable study identifying eh mechanisms underlying the social touch. There are a couple of points that the authors can address in the revised version:

·       Please explain the timing effect: is there any effect on how long this should last for an optimal effect? has it been tested before? and also the interval of the touch? continuous versus on-off style. Please elaborate.

·       Since all participants here are healthy and the pain is the experimental state here, how would the author see the situation for pain patients? Would that be a similar effect or some diversity is expected? For example in chronic pain patients, does the social touch create a similar, higher, or lower effect? Since this is a non-invasive method, it can be used for soothing purposes? Please elaborate.

·       Has this phenomenon been tested to identify if any age or gender difference exists? For example in pediatrics versus adults or women than men? Please elaborate.

·       Is this phenomenon body-region dependent? lower limbs, upper limbs, head, face, abdominal, back, etc. Please elaborate.

Author Response

R1:

In this study, human-human gentle skin stroking is effective in activating CT afferents and reducing pain which is most likely through neural processes of the descending pain pathway. The researchers developed standardized human-human touch gestures for signaling attention and calming. The attention gesture was by tapping and the calming gesture was by slow stroking of the skin, where the attention gesture was ineffective and calming gesture effective in activating CT-afferents. The investigators studied touch analgesia using functional brain imaging. The slow stroking gesture reduces pain through a mechanism involving the insular cortex and periaqueductal gray. This study showed that non-invasive social somatosensory stimulus could reduce experimental pain. The study group was all healthy volunteers with no pain. Based on the findings, the researchers propose that the insular cortex and the PAG are involved. However, other for other potential cortical mechanisms further studies are required.

This is a valuable study identifying eh mechanisms underlying the social touch. There are a couple of points that the authors can address in the revised version:

  • Please explain the timing effect: is there any effect on how long this should last for an optimal effect? has it been tested before? and also the interval of the touch? continuous versus on-off style. Please elaborate.

Answer: We agree that this is a very interesting question, which should be investigate more systematically. The timing of our experiment was based on early pilot testing which showed a stronger effect the closer in time the touch and pain events occured. Touch duration were based on studies using the same touch-paradigm commonly used in our lab during fMRI. We added this information to the methods section: “Duration, timing, and location of the touch was based on pilot testing and touch- and pain-task commonly used in our lab [19,27-29], we did however not systematically identify the most effective timing and distance between touch- and pain-stimuli.”

  • Since all participants here are healthy and the pain is the experimental state here, how would the author see the situation for pain patients? Would that be a similar effect or some diversity is expected? For example in chronic pain patients, does the social touch create a similar, higher, or lower effect? Since this is a non-invasive method, it can be used for soothing purposes? Please elaborate.

Answer: The effect of gentle touch in patients suffering from chronic pain is an interesting topic in its own right. Depending on the cause of the pain, there are indications that the qualitative perception of gentle touch changes, potentially involving endogenous opioids (Case et al., Touch Perception Altered by Chronic Pain and Opioid Blockade). However, the effect of our touch model on chronic pain is outside the scope of the present study, which aimed at laying the base understanding of touch-pain-interactions. Chronic pain studies would be an interesting follow-up experiment. We added this to the discussion: “The role of social touch in pain analgesia and the potential involvement of different neuroendocrine systems warrant further investigation, specifically with regard to potential applications in chronic pain conditions.”

  • Has this phenomenon been tested to identify if any age or gender difference exists? For example in pediatrics versus adults or women than men? Please elaborate.

Answer: With aging there is a naturally occurring degeneration of peripheral nerves. A review by our McIntyre et al. (2021) describes the various sensory mechanisms and how the tactile resolution deteriorates. Interestingly enough, Sehlstedt et al. (2016) reports that older participants rated social touch as more pleasant than did younger participants. Russo et al. (2020) also report women generally reporting a more pleasant experience from touch than men. Considering our small sample size, we were unfortunately not able to investigate age- or gender-related effects in our findings. We added this to the future directions in the discussion: “There are several open questions that future lines of research need to investigate in more detail. First, our sample was too small to investigate gender- or age-related effects, which might be of special importance when considering potential applications that would utilize social touch for analgesia in pain conditions.”

  • Is this phenomenon body-region dependent? lower limbs, upper limbs, head, face, abdominal, back, etc. Please elaborate.

Answer: From our early pilot testing and our lab’s unpublished data, there seems to be a proximity effect of the touch and pain regions. We applied gentle touch to the ventral surface of the right forearm while we administered the same kind of mechanical pain using a hand-held algometer to the ipsi- and contralateral thenar muscles, as well as the ipsilateral tibialis anterior. The greatest reduction in pain was seen when the site of pain was the most proximal to the area which was touched. We included this point in the methods as follows: “Duration, timing, and location of the touch was based on pilot testing and touch- and pain-task commonly used in our lab [19,27-29], we did however not systematically identify the most effective timing and distance between touch- and pain-stimuli.”

Reviewer 2 Report

In this paper, the authors investigated skin-to-skin touching (stroking or tapping), that can be followed by low and high pain stimuli. They collected behavioral pain-ratings and fMRI measures during the tactile stimulation. Results showed a significant reduction of self-reported pain ratings when the stroking gesture preceded the painful stimulation. At the fMRI level, individual pain ratings were correlated with insula activation, but the activation was not higher for stroking than for tapping in any brain area during the stimulation period. This was true only during the evaluation period, whereby stronger activation in the periaqueductal gray was observed after calming touch compared to after tapping touch. The authors concluded that human-human gentle skin stroking reduced pain through neural processes involving CT-afferents and the descending pain pathway. 

I have several methodological comments:

-       The authors report that “Data cannot be made publicly available due to missing participant consent”. I am not sure that this statement is acceptable from an “open science “perspective.

-       The authors report that “Behavioral data were not normally distributed and analyzed in SPSS (IBM) using non-parametric tests”: please specify this information (values and significance of the normality tests). 

-       I cannot understand why the authors did not apply a non-parametric ANOVA for analyzing their behavioral data (e.g., Friedman test). In this way, they could have run a 2x2 factorial model (factor1 - stimulation: stroking vs tapping; factor 2 - pressure: high vs. low – see also my comment on fMRI analyses).

-       Why using a 6mm full width at half maximum instead of the standard 8mm filter proposed by SPM default settings?

-       I cannot understand why the authors limited their analyses (second and third model) to insula and PAG. Pain perception is associated with a wider neural network (e.g., pain matrix). 

I can't entirely agree with this ROI approach.

-       Second-level fMRI analyses: I strongly suggest running a 2x2 factorial model with (stimulation: stroking vs tapping; pressure: high vs. low). In this way, they could have modeled both main effects and interactions. A series of t-tests cannot be considered the appropriate statistical approach to test such effects.

-       Please, report the effect size of the behavioral results.

-       In Figure 5, the authors highlighted “More activation in the PAG after calming touch than after tapping touch during the rating period, [-3 -28 -16], p<0.001”. However, there are several activations well outside the PAG. I cannot understand how it is possible since they conducted a ROI-based analysis. Having said that, why not discuss these (significant?) activations?

Author Response

R2:

In this paper, the authors investigated skin-to-skin touching (stroking or tapping), that can be followed by low and high pain stimuli. They collected behavioral pain-ratings and fMRI measures during the tactile stimulation. Results showed a significant reduction of self-reported pain ratings when the stroking gesture preceded the painful stimulation. At the fMRI level, individual pain ratings were correlated with insula activation, but the activation was not higher for stroking than for tapping in any brain area during the stimulation period. This was true only during the evaluation period, whereby stronger activation in the periaqueductal gray was observed after calming touch compared to after tapping touch. The authors concluded that human-human gentle skin stroking reduced pain through neural processes involving CT-afferents and the descending pain pathway. 

I have several methodological comments:

-       The authors report that “Data cannot be made publicly available due to missing participant consent”. I am not sure that this statement is acceptable from an “open science “perspective.

Answer: We agree that public sharing of data is of high importance for open science. However, participant privacy and consent are to be treated as at least equally important. This study had received ethics during a phase, when we did not yet include a paragraph on data sharing in the consent form, it is therefore not possible for ethical reasons to share the data publicly.

-       The authors report that “Behavioral data were not normally distributed and analyzed in SPSS (IBM) using non-parametric tests”: please specify this information (values and significance of the normality tests). 

Answer: We added this information to the statistical analysis section of the methods part: “(Shapiro-Wilk-test for low pressure ratings: 0.92, p=0.028, for high pressure ratings: 0.91, p=0.016)”.

-       I cannot understand why the authors did not apply a non-parametric ANOVA for analyzing their behavioral data (e.g., Friedman test). In this way, they could have run a 2x2 factorial model (factor1 - stimulation: stroking vs tapping; factor 2 - pressure: high vs. low – see also my comment on fMRI analyses).

Answer: We thank the reviewer for spotting this omission. We added the overall ANOVA (Friedman test) and kept the Wilcoxon test for the post-hoc comparisons: “During the touch-pain-task, there was an overall statistically significant difference in perceived pain depending on type of pressure and type of touch: χ2(3) = 50.6, p < 0.001. Post-hoc comparisons revealed that…”

-       Why using a 6mm full width at half maximum instead of the standard 8mm filter proposed by SPM default settings?

Answer: This smoothing setting has been used by us and others regularly. A slightly lower smoothing kernel will be beneficial to detect smaller clusters, which is especially important when interested in small brain regions like amygdala or PAG.

-       I cannot understand why the authors limited their analyses (second and third model) to insula and PAG. Pain perception is associated with a wider neural network (e.g., pain matrix). 

I can't entirely agree with this ROI approach.

Answer: We agree that there are many regions that have been implicated in pain processing before. We did however define these a priori regions of interest based on our specific topic of pain-touch-interaction. The insula is well-known to be involved in both pain-processing as well as affective (slow stroking) touch. The PAG is considered a major hub of the descending pain control. We decided to define these ROIs a priori in order to not compare too many regions, but to be able to detect effects that might be too weak to detect at the whole brain level, since we expected the brain signatures related to touch-pain-interaction to potentially not be detectable at a stringent threshold at the whole brain level.

-       Second-level fMRI analyses: I strongly suggest running a 2x2 factorial model with (stimulation: stroking vs tapping; pressure: high vs. low). In this way, they could have modeled both main effects and interactions. A series of t-tests cannot be considered the appropriate statistical approach to test such effects.

Answer: We agree that a repeated-measures ANOVA is appropriate for investigating the interaction of pressure type and touch type in the first analysis, which we performed. We did not find a significant interaction. We added this to the results section: “Furthermore, there was no significant interactions between the two pressure stimuli with the two types of preceding touch. However, this type of analysis did not take individual pain perception into account.” In the following analyses, we merged high- and low-pressure trials and added the individual ratings as a parametric modulator since we considered the individual perception of the pain intensity more important. Therefore, we are here always only comparing the effect of the parametric modulator, which is done using paired t-test. We clarified this in the methods: “Repeated-measures ANOVA (first level model 1) and paired t-tests (first level models 2 and 3) were used to compare conditions.”

-       Please, report the effect size of the behavioral results.

Answer: We added this information to the behavioral results section: “During the touch-pain-task, there was an overall statistically significant difference in perceived pain depending on type of pressure and type of touch: χ2(3) = 50.6, p < 0.001. Post-hoc comparisons revealed that the pain ratings during high pressure were significantly higher than during low pressure (median high: 27.92 (IQR 31.5), range 3.6-70.8; median low: 8.25 (IQR 18.49), range 0-41.5; Z=-4.36, p<0.001, effect size: 0.9). Overall, stroking preceding the high/low pressure-stimulus led to lower pain ratings than tapping (median stroking: 16.5 (IQR 24.8), median tapping: 22.75 (IQR 23.42), Z=-2.35, p=0.019, effect size: 0.2). This effect was mostly driven by the individual differences in pain ratings when the low-pressure stimuli were preceded by stroking or tapping touch (median stroking: 8.5 (IQR 16.8), mean stroking: 10.6 +-10.2; median tapping: 8.2 IQR 20.2, mean tapping: 13.6 +-13.2; Z=-2.76, p=0.006, effect size: 0.2). While not significant, pain ratings were also lower when the high-pressure stimuli were preceded with stroking compared to tapping (median stroking: 25.5 (IQR 28.1), mean stroking 29.76,+-21.4, median tapping: 27.83 (IQR31.9), mean tapping: 31.9+-21.8, Z=-1.58, p=0.113).”

-       In Figure 5, the authors highlighted “More activation in the PAG after calming touch than after tapping touch during the rating period, [-3 -28 -16], p<0.001”. However, there are several activations well outside the PAG. I cannot understand how it is possible since they conducted a ROI-based analysis. Having said that, why not discuss these (significant?) activations?

Answer: Figure 5 is thresholded at p<0.001 for display purpose as stated, however, the other areas do not survive at a the whole-brain-level when applying a peak-threshold of p(FWE-corrected)<0.05, which is the standard for SPM whole brain level multiple comparison corrections. The activation in the PAG displayed in this figure is significant when correcting for the a priori ROI. We added an explanation to the figure legend: “Additionally visible clusters are not significant at the whole brain level and are not part of the ROI.”